# Adaptive Top-K Algorithm for Medical Conversational Diagnostic Model

**DOI:** 10.3390/e26090740

**Published:** 2024-08-30

**Authors:** Yiqing Yang, Guoyin Zhang, Yanxia Wu, Zhixiang Zhao, Yan Fu

**Affiliations:** Department of Computer Science, Harbin Engineering University, Harbin 150001, China; yangyiqing@hrbeu.edu.cn (Y.Y.); zhangguoyin@hrbeu.edu.cn (G.Z.); 641zzx@hrbeu.edu.cn (Z.Z.); fuyan@hrbeu.edu.cn (Y.F.)

**Keywords:** differential diagnosis systems, Top-K algorithm, reinforcement learning

## Abstract

With advancements in computing technology and the rapid progress of data science, machine learning has been widely applied in various fields, showing great potential, especially in digital healthcare. In recent years, conversational diagnostic systems have been used to predict diseases through symptom checking. Early systems predicted the likelihood of a single disease by minimizing the number of questions asked. However, doctors typically perform differential diagnoses in real medical practice, considering multiple possible diseases to address diagnostic uncertainty. This requires systems to ask more critical questions to improve diagnostic accuracy. Nevertheless, such systems in acute medical situations need to process information quickly and accurately, but the complexity of differential diagnosis increases the system’s computational cost. To improve the efficiency and accuracy of telemedicine diagnostic systems, this study developed an optimized algorithm for the Top-K algorithm. This algorithm dynamically adjusts the number of the most likely diseases and symptoms by real-time monitoring of case progress, optimizing the diagnostic process, enhancing accuracy (99.81%), and increasing the exclusion rate of severe pathologies. Additionally, the Top-K algorithm optimizes the diagnostic model through a policy network loss function, effectively reducing the number of symptoms and diseases processed and improving the system’s response speed by 1.3–1.9 times compared to the state-of-the-art differential diagnosis systems.

## 1. Introduction

With advancements in computing technology and the rapid development of data science, machine learning has been widely applied across various industries, showing great potential, especially in digital healthcare. In 2022, China emphasized the development of intelligent auxiliary diagnosis systems for primary healthcare information systems in its “14th Five-Year Plan” for National Health Informatization [1]. As a central pillar of modern medical services, telemedicine diagnosis aims to conserve medical resources and improve healthcare efficiency by providing doctors with auxiliary diagnoses, effectively addressing the inadequacies of primary healthcare systems [1].

In recent years, with the emergence of conversational diagnosis, experts have shown great interest in using symptom checking (SC) to diagnose potential diseases. Early SC systems were primarily trained to collect relevant evidence and predict the likelihood of a single disease in patients while minimizing the number of questions asked to the patient [2,3]. However, they overlooked that in accurate medical diagnoses, doctors tend to perform differential diagnoses by considering multiple potential diseases without further examination results. This helps explain diagnostic uncertainties and guides doctors on which questions to ask during patient interactions, improving diagnostic accuracy [4,5].

However, in emergency and acute care settings, diagnostic systems must process severe medical conditions quickly and accurately to ensure timely patient treatment [3,6,7,8,9,10]. Due to the consideration of more possible diseases in differential diagnosis systems, the number of interactions with patients increases, thus raising the system’s computational cost [4]. Therefore, improving diagnostic efficiency while ensuring accuracy is a challenge.

To enhance the efficiency and accuracy of telemedicine diagnostic systems, we developed an optimized technique for the Top-K algorithm. The diagnostic system is optimized from two essential aspects: dynamically adjusting the number of symptoms asked of patients and dynamically selecting the top K most likely diseases during the diagnostic process rather than focusing solely on a single disease or all possible diseases. Additionally, the Top-K algorithm improves the performance of the diagnostic model through policy network loss function optimization. This method not only enhances the efficiency of the diagnostic process but also ensures high accuracy (99.81%), reducing the system’s response time by 1.3–1.9 times. The contributions of this study are as follows:**Enhanced diagnostic accuracy**: The Top-K algorithm dynamically selects the top K most likely diseases during diagnosis, effectively simulating the diagnostic logic of real doctors, thereby improving diagnostic accuracy (99.81%).**Efficient utilization of system resources:** By reducing the number of symptoms and pathologies processed by the system, the system’s response time is significantly improved (1.3–1.9 times), achieving efficient utilization of resources.**Robustness and stability of the system:** The Top-K algorithm optimizes diagnostic strategies based on real-time data and the specific needs of cases by continuously monitoring the progress of cases, enabling the system to adapt to various environments and changes, enhancing system robustness and stability.**Improved handling efficiency of severe pathologies:** By dynamically adjusting the K value to prioritize high-risk pathologies, the diagnostic speed and accuracy of severe pathologies are improved, effectively reducing the misdiagnosis rate of serious diseases and enhancing the efficient use of medical resources.

The introduction of the Top-K algorithm not only addresses the core challenges in telemedicine diagnosis but also provides new directions and possibilities for the future development of medical systems. This innovative diagnostic technology, by precisely controlling the number of symptoms and pathologies processed, not only conserves valuable medical resources but also improves diagnostic efficiency, providing healthcare providers with an efficient and reliable tool to meet the increasingly complex medical needs.

This research presents an Adaptive Top-K Algorithm (TPKA) designed for medical conversational diagnostic systems. The related work section reviews existing telemedicine diagnostic systems and the application of the Top-K algorithm in the medical field, emphasizing the need for enhanced diagnostic accuracy through reinforcement learning. The methodology section details the design and implementation of TPKA, focusing on its dynamic adjustment mechanism, which optimizes the number of symptoms and diseases considered during diagnosis, thereby improving system efficiency and reducing computational overhead. The experiments validate TPKA’s superior performance in differential diagnosis tasks compared to existing systems, demonstrating its effectiveness in enhancing diagnostic efficiency and handling severe pathologies. The discussion highlights TPKA’s potential to improve diagnostic accuracy and efficiency in conversational systems, acknowledges the study’s limitations, and suggests future research directions, such as using more representative datasets and integrating physician feedback to further optimize system performance.

## 2. Related Work

### 2.1. Telemedicine Diagnostic Systems

Several approaches have been developed for evidence acquisition and automatic diagnosis, which gather symptom information through patient interactions and assist doctors in disease diagnosis. For instance, Kao et al. [2] implemented a context-aware HRL method to enhance symptom-checking (SC) accuracy compared to traditional systems by conducting a limited number of queries. Wei et al. [3] introduced a reinforcement learning-based conversational diagnostic system using a pediatric medical dataset from an online medical community in China, improving the accuracy of automatic diagnosis by collecting patient symptoms through dialogues. Considering differential diagnoses instead of single pathology has the added advantage of addressing inherent diagnostic uncertainties and errors, mainly when decisions rely solely on patient interactions without medical examinations or tests. Therefore, the capability to predict differentials rather than actual pathologies is vital for gaining doctors’ trust in the model. Fansi et al. [4] proposed a reinforcement learning-based conversational system for differential diagnosis, mimicking doctors’ diagnostic processes by collecting symptoms from patients and considering as many potential diseases as possible to improve diagnostic accuracy and handle complex situations. However, this model accounts for the possibility of all diseases, resulting in substantial redundant computation. These studies offer new insights and methods for automating differential diagnosis, enhancing the accuracy of medical diagnosis.

### 2.2. Top-K Algorithm

The Top-K algorithm is a critical data selection tool extensively used in various domains, including recommendation systems, query systems, and scientific research. It optimizes decision-making and improves performance efficiency in deep learning and reinforcement learning. In reinforcement learning environments, the Top-K algorithm is particularly beneficial in balancing exploration and exploitation in large action spaces by prioritizing actions with high potential value, thereby reducing computational overhead [11]. Furthermore, this algorithm improves decision-making strategies by selectively learning from crucial experiences in the experience replay buffer [12]. For computational load optimization, the Top-K algorithm enhances learning efficiency in processing high-dimensional complex data by reducing the problem’s dimensionality [12,13]. In distributed learning, it significantly lowers communication burdens and speeds up model convergence by selecting the top K most important gradients for transmission in each iteration [14]. By identifying top-K actions or strategies based on reward potential, novelty, or error, learning algorithms can use high-impact decisions to optimize their strategy more effectively [11,15]. Focusing on critical actions or items that could enhance recommendation performance reduces computational burdens, allowing the model to operate efficiently even in resource-limited environments and addressing resource consumption issues when handling large datasets [15].

In telemedicine diagnostics, the Top-K algorithm has been successfully utilized in symptom screening and the diagnosis of complex diseases by identifying key symptoms most likely indicative of specific diseases, assisting doctors in making more accurate diagnostic decisions. For example, applying deep learning techniques in data feature extraction has demonstrated the ability to accurately identify key diagnostic indicators through the Top-K algorithm [16]. In medical image processing, the Top-K method optimizes the model training process and improves diagnostic accuracy by focusing on image areas with significant pathological features [17]. Intelligent consultation systems also employ the Top-K algorithm, enhancing primary screening efficiency and accuracy by prioritizing the most critical symptoms and patient historical data [18]. This approach streamlines the diagnostic process and enhances telemedicine services’ overall efficiency and accuracy.

## 3. Methodology

This section comprehensively introduces the design, implementation, and impact of the Top-K algorithm (TPKA) in conversational diagnostic systems. We first present the TPKA algorithm (Section 3.1), detailing its implementation and role in conversational diagnostic systems through a thorough explanation of its dynamic adjustment mechanism (Section 3.1.1) and specific algorithm design and implementation (Section 3.1.2). Next, we discuss how TPKA improves system efficiency by filtering the Top-K symptoms and pathologies, reducing the options to be considered. Additionally, since the varying K value changes the number of pathologies and symptoms considered, TPKA significantly impacts the reward mechanisms in the conversational diagnostic system, such as weights and priorities, which we will discuss in Section 3.2. Finally, Section 3.3 introduces how TPKA affects the training process of the conversational diagnostic system.

### 3.1. TPKA

In this conversational diagnostic system, we follow the dual policy proposed by Arsène et al. [4], which consists of two key components: the evidence acquisition module (q) and the differential generation module (pi). The q-value represents the inquiry value or priority of the current patient’s symptoms, while the p-value represents the probability distribution of the patient’s potential pathologies. These values are predicted by a reinforcement learning model that receives medical history and real-time conversational input as features. The system uses the q-value to determine the following symptoms to inquire about based on the information collected in the current conversation and existing medical knowledge. By asking about the symptoms with the highest q-value, the system can more effectively gather the most crucial information for diagnosis. The pi-value, on the other hand, is directly related to disease prediction. Based on the p-value, the system evaluates the patient’s likelihood of having certain diseases and ranks these pathologies accordingly.

TPKA influences these two values, meaning the system chooses the most likely K symptoms to inquire about (for q-values) or predicts the most likely K pathologies (for pi-values) at each step. This approach improves diagnostic efficiency and makes the diagnostic process more focused and precise. The core principle of TPKA is to optimize the inquiry and diagnosis process through dynamic adjustment. This algorithm considers multiple parameters, such as the quality of responses in the conversation, the urgency of symptoms, and the severity of pathologies. This selection reduces unnecessary inquiries while ensuring the rational use of medical resources, which is especially important in resource-limited situations. Furthermore, the system can continuously optimize its performance by utilizing a feedback loop, enhancing diagnostic accuracy and efficiency.

#### 3.1.1. Dynamic Adjustment Mechanism

This section explicitly explains the dynamic adjustment mechanism of TPKA in the conversational diagnostic system, illustrating the role of TPKA in the system and how it affects the conversational system. As the diagnostic conversation progresses, the algorithm dynamically adjusts the number of symptom inquiries (for q-values) and pathology predictions (for pi-values) to ensure the model can focus on handling the most likely scenarios at each stage. This dynamic adjustment is implemented based on performance metrics such as accuracy, recall rate, and other clinical outcome indicators (diagnosis data), as expressed in Equation (Equation 1).
(1)K(t)=f(t,PerformanceMetrics,DiagnosesData)
where *f* is a function that dynamically adjusts K(t) based on the time step *t*, performance metrics (such as accuracy and recall rate), and historical diagnostic data. Performance metrics can include Top-K accuracy, among others, and historical diagnostic data, which refers to the evidence collected by the model in previous diagnostic steps. This dynamic adjustment mechanism ensures the model can focus on the most likely symptoms and pathologies at each stage, improving diagnostic accuracy and efficiency.

Figure 1 shows an automated conversational diagnostic system incorporating TPKA. TPKA is a key component of the conversational diagnostic system, dynamically adjusting the focus according to the current conversation context and selecting the most critical symptoms and diseases to inquire about and predict. In this system, deep learning architectures are employed to achieve automated evidence acquisition and differential diagnosis, emulating the diagnostic process of a physician. The system’s design draws from multiple prior studies, effectively integrating reinforcement learning and deep neural networks to assess and make decisions regarding a patient’s condition. A key component is the Encoder Network, a multilayer perceptron (MLP) responsible for processing various data collected from the patient, including symptoms, medical history, and demographic information. The task of the Encoder Network is to transform these raw input data into a latent representation, a high-dimensional feature vector that captures the essential characteristics and complex relationships within the input data. This latent representation is crucial for other parts of the model, as it provides a condensed, information-rich foundation for both the Classifier Network and the Policy Network. The Classifier Network utilizes the latent representation generated by the Encoder Network to predict the differential diagnosis of the patient, inferring a list of potential diseases based on the current evidence. Meanwhile, the Policy Network leverages this latent representation to determine further actions, such as querying additional symptom information or terminating the interaction.

In the conversational diagnostic system, TPKA is used to filter the most critical $K_{q}$ symptoms (i.e., q-values) and the most likely $K_{pi}$ diseases (i.e., pi-values), adjusting the K values through interaction with the environment.

Specifically, for q-values, which represent the symptoms to be inquired from the patient, the Top-K algorithm ensures that the system focuses on the most critical symptoms at a time, making the diagnostic process both efficient and targeted. This selection is based on real-time performance metrics such as recall rate, precision, and environment feedback. Additionally, the system adjusts the K value based on historical data and the patient’s response to ensure a focus on the symptoms most likely to occur at different stages.

For pi-values, representing the system’s predicted disease probability distribution, TPKA helps the system focus on the most likely diseases, ignoring less probable ones. This is also a dynamic process where the system adjusts its focus based on the accuracy of the current diagnostic situation, considering historical data and predictive uncertainty. By changing the number of diseases considered, the system maintains a balance between certainty and exploration, ensuring that key pathologies are not overlooked while avoiding resource wastage on less probable ones. Additionally, through empirical observation [4], the reward weights affect the system’s performance at different stages (exploration or confirmation), so TPKA automatically adjusts the reward weights to balance exploration and confirmation at different stages to improve diagnostic accuracy and efficiency. Overall, TPKA provides a dynamic learning and adaptation mechanism for conversational diagnostic systems, ensuring that the system can inquire about symptoms and predict diseases in the most rational way, ultimately enhancing diagnostic accuracy and efficiency. By finely adjusting q-values and pi-values, the algorithm offers strong support for decision-making in uncertain and variable diagnostic scenarios.

#### 3.1.2. Algorithm Implementation

This section details the implementation of TPKA, which continuously adjusts the K value (i.e., the number of diagnostic items considered at a time and the number of most likely diseases considered) to focus attention and resources on the most valuable options while finding a balance between exploration and confirmation. It ensures that the system can promptly respond to patient feedback during the diagnostic process and adjust its focus between accuracy and exploration through real-time performance evaluation and weight adjustment. The performance metrics involved in the algorithm are as follows:**Recall** This metric measures the model’s ability to capture positive samples. In medical diagnosis, a high recall rate means that more actual positive cases can be identified, reducing the risk of missed diagnoses. If the recall rate is low, consider increasing the K value, i.e., including more possible diseases in the selection of pi values to ensure that critical cases are not overlooked.**Precision** This metric measures the accuracy of the model’s predictions. If the precision is low, it may be necessary to reduce the K value and focus on higher probability diseases to avoid excessive misdiagnosis.**Top-K Accuracy (GTPA@K)** This metric measures whether the top K items of differential diagnosis predicted by the agent include the patient’s true pathology, as referenced in [4].

Improving recall often sacrifices precision, and vice versa. A high recall rate leading to many irrelevant symptoms being inquired may require lowering the recall threshold; if high precision results in missing important symptoms, the precision threshold may need to be lowered. By monitoring these performance metrics in real-time, the system can dynamically adjust the K value according to the current diagnostic situation to balance accuracy and efficiency. The Top-K accuracy metric can monitor the accuracy of the K pathologies predicted by TPKA in real-time and provide feedback for adjusting the K value.

Algorithm 1 focuses on the q-values for symptom inquiries. By monitoring precision and recall, this algorithm determines the number of critical symptoms to be inquired about during the diagnostic conversation. If the system’s diagnostic precision is high, it reduces the number of symptoms inquired to avoid overburdening the patient. Conversely, if the recall rate is low, the system will inquire about more symptoms to increase diagnostic coverage.
**Algorithm 1** Adaptive Top-K Algorithm for Symptom Inquiry (q-values).1:**Input:** Symptom q-values, Performance Thresholds2:**Initialize:** 
$K_{q}\leftarrow K_{\mathrm{initial}}$3:**while** session is active **do**4:    Calculate current performance metrics (Precision, Recall)5:    **if** Recall < Recall_Threshold **then**6:        $K_{q}\leftarrow \min\left(K_{q}+\Delta K,K_{\max}\right)$        ▹ Increase K if underperforming7:    **else if** Precision > Precision_Threshold **then**8:        $K_{q}\leftarrow \max\left(K_{q}-\Delta K,K_{\min}\right)$        ▹ Decrease K if overperforming9:    **end if**10:  Select top-$K_{q}$ symptoms based on q-values11:  Ask patient about selected symptoms12:  Update q-values based on patient feedback13:  Reevaluate performance metrics14:**end while**15:**Output:** Inquired symptoms

Algorithm 2 focuses on the pi-values (probability distribution of pathologies) for disease prediction. Each iteration calculates the pi-values and selects the Top-K pathologies with the highest probabilities as prediction results. The pi-values are updated based on the information collected during the interaction. Similarly, the K value $K_{pi}$ is dynamically adjusted based on performance metrics. Depending on different indicators of Top-K accuracy, the algorithm adjusts the number of diseases considered to ensure focus on the most likely diseases. In this way, the system neither misses critical diagnoses nor overemphasizes less likely diseases.
**Algorithm 2** Adaptive Top-K Algorithm for Disease Prediction (pi-values).1:**Input:** Disease pi-values, Performance Metrics (GTPA@K)2:**Initialize:** 
$K_{\pi }\leftarrow K_{\mathrm{initial}}$3:**while** session is active **do**4:    Calculate current GTPA@K5:    **if** GTPA@K is low **then**6:        $K_{\pi }\leftarrow \min\left(K_{\pi }+\Delta K,K_{\max}\right)$        ▹ Adapt K based on performance7:    **end if**8:    Select top-$K_{\pi }$ diseases based on pi-values9:    Update diagnosis based on additional tests or doctor reviews10:  Reevaluate GTPA metrics11:**end while**12:**Output:** Predicted diseases

Algorithm 3 is the automatic adjustment strategy for reward weights in TPKA. It continuously adjusts the $K_{q}$ value (i.e., the number of diagnostic items considered at one time) to find a balance between exploration and confirmation, and determines whether the current diagnostic stage is exploration or confirmation. Through real-time performance evaluation and weight adjustment, the algorithm ensures a timely response to patient feedback during the diagnostic process, and adjusts the focus between accuracy and exploration. If high information uncertainty is found in the early stages of the conversation, $K_{pi}\left(t\right)$ may be set larger to allow the model to explore more potential pathologies. As the conversation progresses and data accumulates, $K_{pi}\left(t\right)$ will gradually decrease, and the model’s focus will become more concentrated on the most likely pathologies.
**Algorithm 3** Dynamic Adjustment Implementation of Adaptive Top-K Algorithm.1:**Input:** Data stream *D*, time window *T*, performance metrics *P*, initial K value $K_{0}$2:**Initialize:** t←0, $K\leftarrow K_{0}$, exploration weight $w_{Ex}\left(t\right)\leftarrow 1$, confirmation weight $w_{Co}\left(t\right)\leftarrow 0$3:**Define:** Performance threshold τ4:**while** data stream *D* not finished **do**5:    t←t+16:    Receive current data $d_{t}$ from *D*7:    Update performance metrics P(t) based on $d_{t}$ and previous data8:    K←ADJUSTK(t,P(t),K)        ▹ Dynamically adjust K value9:    **if** P(t)<τ **then**10:        $w_{Ex}\left(t\right)$ increases         ▹ Increase exploration weight11:        $w_{Co}\left(t\right)$ decreases         ▹ Decrease confirmation weight12:    **else**13:        $w_{Ex}\left(t\right)$ decreases         ▹ Decrease exploration weight14:        $w_{Co}\left(t\right)$ increases         ▹ Increase confirmation weight15:    **end if**16:    Select Top-K elements based on $w_{Ex}\left(t\right)$, $w_{Co}\left(t\right)$, and current *K*17:    Process and output the selection result18:**end while**

By combining these three algorithms, we achieve a powerful and flexible dialogue diagnostic system. It can dynamically adjust diagnostic strategies based on real-time data and performance metrics, ensuring that each diagnostic step is purposeful and data-driven. This adaptability is key to efficient and accurate diagnosis, not only improving patient experience but also maximizing the utilization efficiency of medical resources. Through the operation of these algorithms, the system can accurately predict symptoms and diseases, providing strong decision support for doctors.

### 3.2. Reward Mechanism and TPKA

The dynamic adjustment mechanism of TPKA is combined with different types of rewards to further improve the performance and efficiency of the diagnostic system. The reward mechanism is based on the dynamic adjustment theoretical framework, which adjusts in real-time according to system performance, enhancing the accuracy and reliability of the system in handling the most likely symptoms and pathologies.

#### 3.2.1. Exploration Reward

The exploration reward ($R_{\mathrm{Ex}}$) aims to encourage the diagnostic system to broadly consider potential diagnoses in the early stages of interaction, dynamically adjusting the depth and range of symptom ($K_{q}$) and pathology ($K_{pi}$) exploration. This exploration not only improves the system’s ability to identify various possibilities but also optimizes the performance of the diagnostic system across multiple dimensions. The calculation formula for the exploration reward is shown in Equation (Equation 2).
(2)$$R_{\mathrm{Ex}}\left(s_{t},a_{t},s_{t+1},K_{q},K_{pi}\right)={⊮}_{s_{t+1}\neq s_{⊥}}w_{\mathrm{Ex}}\left(t,K_{q},K_{pi}\right)\times D\left(t,K_{q},K_{pi}\right)$$
where $D(t,K_{q},K_{pi})$ represents the depth of exploration, calculated as the sum of the Jensen–Shannon divergence (JSD) of symptoms and pathologies, specifically defined in Equation (Equation 3).
(3)$$D\left(t,K_{q},K_{pi}\right)=\sum_{i=1}^{K_{q}\left(t\right)}\mathrm{JSD}\left(b_{\mathrm{el}}^{t}\left[i\right],b_{\mathrm{el}}^{t+1}\left[i\right]\right)+\sum_{i=1}^{K_{pi}\left(t\right)}\mathrm{JSD}\left(b_{\mathrm{el}}^{t}\left[i\right],b_{\mathrm{el}}^{t+1}\left[i\right]\right)$$
The exploration weight function $w_{\mathrm{Ex}}\left(t,K_{q},K_{pi}\right)$ adjusts the importance of exploration, dynamically adjusting based on the time step *t* and the total amount of symptom and pathology exploration $\left(K_{q}+K_{pi}\right)$, expressed as shown in Equation (Equation 4).
(4)$$w_{\mathrm{Ex}}\left(t,K_{q},K_{pi}\right)=\frac{1}{1+e^{-\alpha (t-\beta \left(K_{q}+K_{pi}\right))}}$$
Here, α and β are adjustment factors used to control the rate of change in weights over time and exploration depth, ensuring that the system adjusts its focus as needed at different stages, thus effectively balancing exploration and certainty in the diagnostic process.

TPKA plays a crucial role in the exploration reward mechanism. By dynamically embedding the range and depth of exploration ($K_{q}$ and $K_{pi}$) into the system, it not only enables the system to broadly consider various potential diagnoses in the early stages of diagnosis but also optimizes the system’s performance under different scenarios. This flexible adjustment strategy enhances the system’s ability to identify complex symptoms and pathologies while ensuring the comprehensiveness and depth of the diagnostic process. In this way, TPKA prompts the model to more effectively explore and evaluate symptoms, ensuring that the correct diagnostic path can be selected from a wider range of possibilities.

#### 3.2.2. Confirmation Reward

The confirmation reward is achieved by reinforcing the model’s confidence in its predictions of the most likely pathologies. This mechanism dynamically adjusts the K value, i.e., the number of considered pathologies, based on real-time performance feedback, ensuring an optimal balance between accuracy and resource allocation. The core of the confirmation reward is calculating the change in cross-entropy loss, an indicator of the deviation between the model’s predictions and the actual pathologies. A reduction in cross-entropy loss indicates that the model is getting closer to the true pathologies with successive diagnostic steps, thus reducing cross-entropy is a key objective for model optimization.

Equation (Equation 5) details the dynamic change in cross-entropy loss and its application in the reward mechanism.
(5)$$\Delta \mathrm{CE}\left(t,K_{pi}\right)=\gamma \cdot \mathrm{CE}\left(b_{\mathrm{el}}^{t+1}\left[1:K_{pi}\left(t\right)\right],y\left[1:K_{pi}\left(t\right)\right]\right)-\mathrm{CE}\left(b_{\mathrm{el}}^{t}\left[1:K_{pi}\left(t\right)\right],y\left[1:K_{pi}\left(t\right)\right]\right)$$
Here, ΔCE captures the change in cross-entropy from one time step to the next, reflecting diagnostic progress or regression. If cross-entropy decreases, it indicates that the model’s predictions are becoming more accurate and reliable.

Additionally, the design of the confirmation weight $w_{\mathrm{Co}}\left(t,K_{pi}\right)$ is to adjust the model’s sensitivity to confirmatory information at different diagnostic stages. The weight function is expressed in Equation (Equation 6).
(6)$$w_{\mathrm{Co}}\left(t,K_{pi}\right)=\max\left(0,1-\frac{t}{T-\gamma K_{pi}}\right)$$
where *T* represents the preset duration of the entire diagnostic dialogue, and γ is an adjustment factor that determines the rate at which the weight decreases with time and the number of considered pathologies. This design ensures that in the early stages of diagnosis, the model has greater flexibility to explore multiple pathologies, while as the diagnosis approaches its end, the model focuses more on the most likely pathologies, thereby improving diagnostic accuracy and efficiency.

This mechanism exemplifies the critical role of the Top-K algorithm in balancing diagnostic accuracy with resource efficiency by purposefully adjusting the range of considered pathologies to optimize model performance.

#### 3.2.3. Severity Pathology Reward

The severity pathology reward ($R_{\mathrm{Sev}}$) mechanism is an important strategy to ensure that critical and potentially threatening pathologies are prioritized. This reward mechanism dynamically adjusts the focus related to severe pathologies to optimize resource allocation and attention during the diagnostic process.

The severity pathology reward reflects changes in the severity of pathologies and is calculated using Equation (Equation 7).
(7)$$R_{\mathrm{Sev}}\left(s_{t},a_{t},s_{t+1},K_{pi}\right)={⊮}_{s_{t+1}\neq s_{⊥}\wedge {\mathrm{SevOut}}^{t+1}\neq {\mathrm{SevOut}}^{t}}\left(\gamma_{\mathrm{SevOut}}^{t+1}-{\mathrm{SevOut}}^{t}\right)$$
In this equation, ${⊮}_{s_{t+1}\neq s_{⊥}\wedge {\mathrm{SevOut}}^{t+1}\neq {\mathrm{SevOut}}^{t}}$ is an indicator function that triggers the reward when there is a change in the severity pathology output between successive diagnostic steps. Here, $\gamma_{\mathrm{SevOut}}^{t+1}-{\mathrm{SevOut}}^{t}$ represents the change in the severity pathology state from one time step to the next, serving as a direct measure of the severity change.

TPKA’s application in this mechanism is crucial. It ensures that the system can identify and prioritize pathologies with high risk and severe consequences during the diagnostic process, aligning diagnostic decisions with the clinical judgment of doctors. By doing so, TPKA helps the model more effectively allocate diagnostic resources, ensuring that the focus remains on pathologies that are most likely to impact the patient’s health.

#### 3.2.4. Classification Reward

The classification reward ($R_{\mathrm{Cl}}$) is a key evaluation mechanism activated at the end of the medical diagnostic dialogue, aimed at ensuring the system’s pathology predictions are highly consistent with the actual conditions. This reward mechanism is calculated using Equation (Equation 8).
(8)$$R_{\mathrm{Cl}}\left(s_{t},a_{t},s_{t+1},K_{pi}\right)={⊮}_{s_{t+1}=s_{⊥}}\left(-\sum_{i=1}^{K_{pi}\left(t\right)}\mathrm{CE}\left(b_{\mathrm{el}}^{t}\left[i\right],y\left[i\right]\right)+w_{\mathrm{si}}\frac{\mathrm{SevInt}}{\mathrm{Sevy}}\right)$$
where ${⊮}_{s_{t+1}=s_{⊥}}$ is an indicator function indicating that the dialogue has ended, at which point the classification reward is calculated. $\mathrm{CE}(b_{\mathrm{el}}^{t}\left[i\right],y\left[i\right])$ represents the cross-entropy loss, measuring the deviation between the model’s prediction for the *i*-th most likely pathology and the actual pathology.

The classification reward calculation also includes a weighting factor $w_{\mathrm{si}}$, which adjusts the impact of severe pathologies, making the model more attentive to clinically severe cases during the evaluation process. $\frac{\mathrm{SevInt}}{\mathrm{Sevy}}$ is a ratio indicating the number of pathologies identified as severe in the predictions to the actual number of severe pathologies, helping to further refine the reward calculation. This classification reward mechanism ensures a strict evaluation of pathology predictions at the end of the dialogue, significantly enhancing the reliability and effectiveness of the diagnostic system. This reward mechanism reflects the key role of the Top-K algorithm in maintaining prediction accuracy and consistency at the end of the diagnosis, optimizing diagnostic results by adjusting the precision of pathology classification.

### 3.3. TPKA in Network Loss Functions

This section elaborates on the application and impact of the policy network and classifier network in TPKA. Firstly, the policy network loss function aims at optimizing the inquiry strategy by dynamically adjusting the number of symptom inquiries, thereby improving the model’s efficiency and accuracy in symptom selection. Secondly, the classifier network loss function focuses on enhancing the accuracy of pathology predictions by adjusting the number of predicted pathologies, ensuring the model operates efficiently under limited computational resources.

#### 3.3.1. Policy Network Loss Function

The loss function of the policy network aims to focus the learning process on the most crucial symptom inquiry strategies. It optimizes the range of symptom selection by dynamically adjusting the Top-K value ($K_{q}$), i.e., the number of symptoms considered at one time. Equation (Equation 9) defines the loss function to ensure that the network output aligns with the expected strategy, thus maximizing the efficiency and accuracy of the entire diagnostic process.
(9)$${\mathrm{Loss}}_{Q}=\frac{1}{2}\sum_{(s_{t},a_{t},r_{t},s_{t+1})\in B}\left[{\left(r_{t}+\gamma \max_{a\in A}Q\left(s_{t+1},a\right)-Q\left(s_{t},a_{t}\right)\right)}^{2}\right]$$
where $r_{t}$ represents the immediate reward obtained after taking action $a_{t}$ in state $s_{t}$, γ is the discount factor for future rewards, representing the current value of future rewards, and $Q(s_{t+1},a)$ is the maximum predicted Q value when taking action *a* in the next state $s_{t+1}$. This temporal difference update method allows the policy network to learn how to select symptoms for inquiry effectively based on the current diagnostic information. By minimizing this loss function, the network learns which symptoms are most critical given the current information and optimizes its inquiry strategy accordingly, achieving effective symptom evaluation and selection. This method not only emphasizes immediate decision feedback but also further optimizes the inquiry process by dynamically adjusting the $K_{q}$ value, making the inquiry of critical symptoms more precise and thereby improving the performance and reliability of the entire medical diagnostic system.

#### 3.3.2. Classifier Network Loss Function

The loss function of the classifier network incorporates the Top-K algorithm to focus on the most probable disease predictions. This method dynamically adjusts the number of interested pathologies $K_{pi}\left(t\right)$, allowing the network to more accurately predict and update critical pathologies. Equation (Equation 10) defines the loss function.
(10)$${\mathrm{Loss}}_{C}=\frac{1}{2}\sum_{(s_{t},a_{t},r_{t},s_{t+1},y)\in B}\left[1_{s_{t+1}=s_{⊥}}\cdot \sum_{i=1}^{K_{pi}\left(t\right)}\mathrm{CE}\left(b_{\mathrm{el}}^{t+1}\left[i\right],y\left[i\right]\right)\right]$$
where $K_{pi}\left(t\right)$ represents the number of pathologies considered at each diagnostic stage, dynamically adjusted based on the latest information obtained by the model and the current diagnostic needs. This method optimizes the model’s resource allocation, ensuring that the most probable pathologies are prioritized under limited computational resources. The integration of the Top-K mechanism allows the classifier to focus more effectively on high-probability pathologies during the diagnostic process, thereby improving overall diagnostic accuracy and efficiency. This adaptive strategy reflects the system’s flexibility in practical operations and its ability to handle complex diagnostic scenarios.

## 4. Experiments

In this section, we provide a detailed overview of the experimental setup (Section 4.1), evaluation metrics (Section 4.2), and results of our study (Section 4.3). Our experiments are designed to validate the effectiveness of the proposed Top-K algorithm (TPKA) in a conversational diagnostic system. We use the DDX Plus dataset, which offers a comprehensive set of differential diagnosis and pathology severity information, to benchmark our approach against several state-of-the-art systems. We compare our method with AARLC [8], Diaformer [19], BED [20], and CASANDE [4], each representing different strategies and techniques in medical diagnostics. The performance of each system is evaluated based on various metrics, including accuracy, recall, F1 score, and diagnostic efficiency. The results highlight the superior performance of TPKA in handling differential diagnosis tasks, optimizing diagnostic processes, and managing severe pathologies. Through a series of quantitative analyses, we demonstrate the potential and practical value of TPKA in modern medical diagnostic systems.

### 4.1. Experimental Setup

We chose the DDX Plus dataset [21] to validate our proposed methods. Unlike previous datasets such as Muzhi [7], HPO [20], and Medline Plus [8], the DDX Plus dataset provides differential diagnosis and pathology severity information, supporting multiple evidence types (binary, categorical, and multiple choice). It includes 49 pathologies and 223 pieces of evidence (110 symptoms and 113 antecedents). Additionally, the dataset is divided into a training subset (over 106 synthetic patients), a validation subset, and a test subset (each containing approximately 140,000 synthetic patients) for comprehensive evaluation.

For baselines, we selected four systems: these baselines were chosen because they demonstrate different approaches and techniques in handling medical diagnostic problems. AARLC showed excellent performance on the Sym CAT dataset, Diaformer displayed competitive results on the Muzhi and DX datasets, while BED is a training-free method that achieved significant results on the HPO dataset. These systems all require adjustments in their ability to handle different types of evidence and differential diagnosis to adapt to the DDX Plus dataset. The CASANDE system is specifically designed to mimic the reasoning process of doctors and focuses on generating differential diagnoses, demonstrating superior performance in handling severe pathologies and differential diagnoses.

Each patient case in the DDX Plus dataset comes with a set of detailed information that plays a key role during the interaction between the model and the patient. This information includes a chief complaint that the model needs to understand at the start of the interaction; a set of additional evidence that the model needs to discover through the inquiry process; and the patient’s differential diagnosis and true pathology. We set a limit of up to 30 turns for these interactions. To optimally evaluate the performance of each model (including baseline models), we fine-tuned the hyperparameters of each model individually on the validation set and reported their performance based on these optimal parameter settings. This process ensures that we can accurately assess and compare the effectiveness and accuracy of different models in handling actual medical diagnostic tasks.

### 4.2. Evaluation Metrics

We used several evaluation metrics to assess the performance of each model. These metrics include accuracy, recall, F1 score, Top-K accuracy (GTPA@K), and mean average precision (MAP). These metrics provide a comprehensive evaluation of the models’ performance in differential diagnosis tasks, offering a thorough assessment of their accuracy, efficiency, and reliability. The evaluation metrics for the adaptive Top-K algorithm (TPKA) include the following aspects, each aimed at evaluating the algorithm’s performance and effectiveness from different angles:
Interaction Length (IL):–Evaluation Objective: This metric evaluates TPKA’s efficiency in reducing the steps required for diagnosis, aiming to improve dialogue efficiency by optimizing the number of symptom inquiries and pathology predictions. A shorter interaction length indicates good algorithm performance.–Evaluation Method: Calculate the average number of interaction steps required to complete a diagnostic task.Positive evidence recall (PER):–Evaluation Objective: Measure the model’s ability to capture evidence relevant to the case, reflecting TPKA’s effectiveness in ensuring diagnostic comprehensiveness.–Evaluation Method: Calculate the proportion of relevant evidence successfully retrieved by the model.Ground Truth Pathological Accuracy (GTPA):–Evaluation Objective: Measure the model’s performance in predicting the correctness of pathologies, with a focus on TPKA’s accuracy in predicting the most likely pathologies.–Evaluation Method: Compare the predicted pathologies with the true pathologies and calculate the accuracy.Differential Diagnosis F1 Score (DDF1):–Evaluation Objective: Evaluate the model’s performance in differential diagnosis, particularly in identifying multiple possible pathologies.–Evaluation Method: Calculate the F1 score of the model’s differential diagnosis predictions.Discharge of Severe Pathologies Harmonic Mean (DSHM):–Evaluation Objective: Reflect the model’s ability to confirm and exclude severe pathologies, assessing TPKA’s effectiveness in handling high-risk pathologies.–Evaluation Method: Calculate the harmonic mean of excluding severe pathologies.Diagnostic Efficiency (DE):–Evaluation Objective: Evaluate the system’s ability to quickly provide accurate diagnostic results.–Evaluation Method: Calculate the average response time required from the start to achieve the correct diagnosis.

### 4.3. Results

This section presents the results and performance of the Top-K algorithm in the conversational diagnostic system. Through a series of detailed evaluations, this chapter reveals the effectiveness and potential advantages of the algorithm in assisting medical diagnosis. We will demonstrate the superior performance of this algorithm from various aspects, including an overview of system performance, model comparison and differentiated performance, detailed evidence and pathology analysis, strategy effectiveness, and handling of severe pathologies. This chapter first details the overall performance of the Top-K conversational diagnostic system in disease prediction and evidence acquisition. The performance of the system in identifying different pathologies and symptoms will be evaluated using quantitative metrics such as accuracy (GTPA), recall (PER), and medical-specific efficiency indicators. According to the data in Table 1, the TPKA system performs excellently across key metrics, particularly in diagnostic efficiency and handling severe pathologies. This indicates that TPKA has significant application value in optimizing diagnostic processes and improving diagnostic accuracy and efficiency. The successful implementation of TPKA demonstrates its great potential and practical value in modern medical diagnostic systems, laying a solid foundation for future research and development of automated medical diagnostic systems.

#### 4.3.1. Differentiated Performance Analysis

The experimental results show that the TPKA system exhibits excellent performance across key metrics. The interaction length (IL) of TPKA is 19.92 steps, slightly longer than Casande’s 19.71 steps but significantly shorter than AARLC’s 25.75 steps and BED’s 17.86 steps. This demonstrates its optimization in inquiry efficiency. The slightly longer interaction length of TPKA might be due to its more detailed exploration phase, ensuring that more critical symptoms are captured. Although this increases the number of inquiry steps, it ultimately contributes to a more accurate diagnosis. Interaction Length (IL) evaluates TPKA’s efficiency in reducing the steps required for diagnosis by calculating the average number of interaction steps needed to complete a diagnostic task. The differential diagnosis F1 score (DDF1) of TPKA is 94.12, close to Casande, and significantly superior to other systems. This indicates its superiority in multi-pathology prediction. TPKA ensures high predictive accuracy and comprehensiveness in multi-pathology situations by optimizing the loss functions of the policy network and classifier network. DDF1 evaluates the model’s performance in differential diagnosis, particularly in identifying multiple possible pathologies, by calculating the F1 score of the model’s differential diagnosis predictions.

#### 4.3.2. Diagnostic Efficiency Analysis

Diagnostic Efficiency (DE) measures the system’s ability to quickly provide accurate diagnostic results. The TPKA system excels in this metric, with a diagnostic time of 15.6 min, significantly lower than other systems, particularly Casande’s 30.3 min. This result indicates that TPKA significantly reduces diagnostic time by optimizing the dynamic adjustment mechanism for symptoms and pathologies, thereby enhancing overall diagnostic efficiency. Ground Truth Pathological Accuracy (GTPA) is 99.81%, slightly higher than Casande’s 99.77%, indicating its advantage in accurately predicting pathologies. GTPA evaluates the model’s performance in predicting the correctness of pathologies by comparing the predicted pathologies with the actual pathologies and calculating the accuracy.

#### 4.3.3. Detailed Evidence and Pathology Analysis

The TPKA system prioritizes the most critical symptoms and predicts the most probable pathologies through the Top-K algorithm, which not only improves diagnostic accuracy but also reduces unnecessary inquiries. In practical applications, TPKA can dynamically adjust the number of symptoms and pathologies, optimizing based on real-time data, allowing the system to more specifically handle the most likely scenarios, thereby enhancing the comprehensiveness and accuracy of the diagnosis. The TPKA’s Query Evidence Recall (PER) is as high as 98.80%, the best among all systems, indicating that TPKA can effectively capture relevant evidence, enhancing diagnostic comprehensiveness. PER evaluates the model’s ability to capture evidence relevant to the case by calculating the proportion of relevant evidence successfully retrieved by the model.

#### 4.3.4. Handling of Severe Pathologies

TPKA performs excellently in handling severe pathologies, with a Discharge of Severe Pathologies Harmonic Mean (DSHM) of 74.10, superior to all other systems. TPKA prioritizes high-risk pathologies by dynamically adjusting the K value, ensuring that critical pathologies can be diagnosed timely and accurately. This prioritization mechanism effectively reduces the missed diagnosis rate of severe diseases and improves the concentrated use of medical resources. DSHM evaluates the model’s ability to confirm and exclude severe pathologies by calculating the harmonic mean of excluding severe pathologies, measuring TPKA’s effectiveness in handling high-risk pathologies.

## 5. Discussion

This study demonstrates the significant potential of the Adaptive Top-K Algorithm (TPKA) in enhancing the efficiency and accuracy of conversational diagnostic systems. By dynamically adjusting the number of symptoms and pathologies considered during diagnosis, TPKA more effectively emulates a physician’s diagnostic logic, resulting in improved diagnostic accuracy (99.81%) and faster response times (1.3–1.9 times). Despite its promising performance, several limitations warrant further investigation. Firstly, TPKA relies on the DDX Plus dataset for training and validation. Although this dataset contains detailed information on differential diagnoses and pathology severity, it may not fully capture the diversity of real-world medical scenarios, potentially leading to data bias. This bias could impair TPKA’s performance when dealing with diverse populations or rare conditions, reducing the system’s generalizability and robustness. Secondly, while TPKA’s dynamic adjustment mechanism optimizes resource allocation during the diagnostic process, it may also introduce algorithmic bias. The algorithm might prioritize based on historical data and common diseases, potentially overlooking atypical symptoms or rare conditions, thus increasing the risk of misdiagnosis or missed diagnosis. Furthermore, TPKA heavily depends on the quality of input data and real-time feedback for its dynamic adjustments. Although this dependency enhances the system’s flexibility and adaptability across various diagnostic stages, it also means that the system’s performance may be adversely affected in cases of low-quality input data or delayed feedback. Therefore, ensuring the quality of input data and the accuracy of real-time feedback is a critical challenge for future research. To address these biases and challenges, future studies should consider using more diverse and representative datasets and incorporating more sophisticated dynamic adjustment strategies to ensure that the algorithm is more sensitive when handling atypical symptoms and rare diseases. Additionally, integrating physicians’ expertise and experience could further enhance the system’s applicability and diagnostic fairness, improving TPKA’s robustness and practical value.

This study establishes the substantial potential of the Adaptive Top-K algorithm in modern medical diagnostic systems, but there remain several avenues for future research. Incorporating more advanced machine learning and reinforcement learning techniques can further optimize TPKA’s dynamic adjustment mechanisms, enhancing its adaptability and robustness. Exploring TPKA’s applicability in other medical fields and scenarios, such as image diagnosis and personalized treatment recommendations, will help validate its broad utility. Moreover, the successful implementation of TPKA requires not only technical improvements but also collaboration with physicians. Researching how to effectively incorporate physician feedback and expertise can further enhance system performance and user experience.

## Figures and Tables

**Figure 1 entropy-26-00740-f001:**
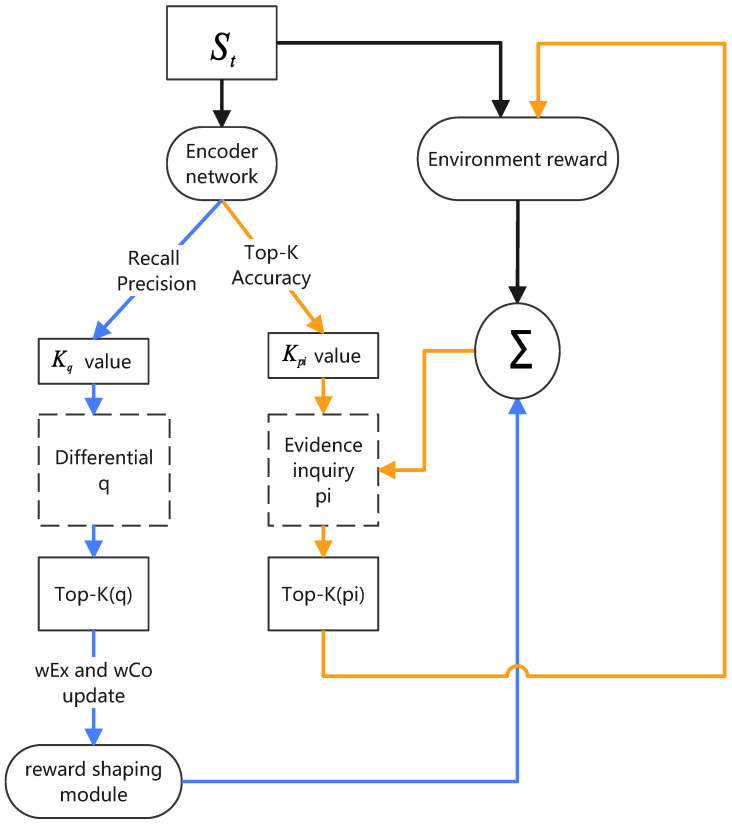
Adaptive Top-K Algorithm. $W_{Ex}$ and $w_{Co}$ represent weight for exploration and classification, respectively. The value of adaptive Top-K is used as a reference to update it, which not only improves the system’s ability to identify possibilities but also optimizes diagnostic system performance in multiple dimensions.

**Table 1 entropy-26-00740-t001:** Evaluation values expressed in percentage (%).

	IL	PER	GTPA	DDF1	DSHM	DE
TPKA	19.92	98.80	99.81	94.12	74.10	15.6
Casande	19.71	98.39	99.77	94.24	73.88	30.3
AARLC	25.75	54.55	99.92	78.24	69.43	22.2
Diaformer	18.41	92.92	99.01	83.3	69.32	24.8
BED	17.86	88.18	67.71	83.69	65.06	20.7

## Data Availability

The original contributions presented in the study are included in the article; further inquiries can be directed to the corresponding authors.

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
