# Peer review of "Adaptive Top-K Algorithm for Medical Conversational Diagnostic Model"

_entropy, 2024, doi:10.3390/e26090740_

Round 1

Reviewer 1 Report

Comments and Suggestions for Authors

The authors try to automate the conventional diagnostic model with adaptive Top-K algorithm. Experts uses the symptom checking method to diagnose potential diseases. The adaptive top-k algorithm enhances the efficiency of diagnostic process and ensures high accuracy with quick response time. The authors done thorough literature survey and identified the research statements.

The authors are suggested to include the following:

1.     The authors need to include reference list and year of publication – missing in the paper

2.     Need to cite the reference number of author name inside the paper – missing

3.     The equations are given, but there is no number and not cited inside the paper – need to include the equation number in the paper

4.     3.3.1 and 3.3.2 are having same title – author need to check and clarify

5.     Query Evidence Recall (PER) – line number 452 – need to check and clarify

Comments on the Quality of English Language

Need to check English grammer

Author Response

We sincerely appreciate the time and effort you have taken to review our manuscript. Your insightful comments have been invaluable in helping us refine and improve the quality of our work. Below, we provide detailed responses to each of your suggestions:

Comments 1: The authors need to include reference list and year of publication – missing in the paper

Response 1: Thank you for your valuable feedback. We have reviewed the reference list and ensured that all relevant year of publication information has been added to the paper.

Comments 2: Need to cite the reference number of author name inside the paper – missing

Response 2: Thank you for bringing this to our attention. We have thoroughly reviewed the citations and ensured that all references now include the appropriate reference numbers following the author's name. We believe this will enhance the clarity and accuracy of the paper.

Comments 3: The equations are given, but there is no number and not cited inside the paper – need to include the equation number in the paper

Response 3: Thank you for your valuable suggestion. We have now added numbers to all equations and cited them appropriately within the paper. We believe these adjustments will improve the overall coherence and readability of the paper.

Comments 4: 3.3.1 and 3.3.2 are having same title – author need to check and clarify

Response 4: We sincerely apologize for the oversight that led to the retention of similar titles for sections 3.3.1 and 3.3.2. This was an unintended result of the drafting process. We have now deleted the content and title of section 3.3.1, ensuring clarity and distinction in the paper. Thank you very much for bringing this to our attention.

Comments 5: Query Evidence Recall (PER) – line number 452 – need to check and clarify

Response 5: Thank you for pointing out this issue. This was indeed a writing error, and it should have been 'Positive Evidence Recall (PER).' This evaluation metric assesses the effectiveness of TPKA in ensuring diagnostic comprehensiveness. We apologize for any confusion this may have caused.

The revised content can be found from page 13 , line 467.

Once again, we would like to express our gratitude for your constructive feedback. We believe that the revisions we have made in response to your comments have significantly strengthened our manuscript. We look forward to any further feedback you may have.

Reviewer 2 Report

Comments and Suggestions for Authors

The topic is both interesting and relevant, as conversational diagnostic systems are actively employed to predict diseases through symptom checking. The paper seeks to improve the efficiency and accuracy of telemedicine diagnostic systems by developing an optimized version of the Top-K algorithm. This enhanced algorithm dynamically adjusts the number of likely diseases and symptoms based on real-time case monitoring, thereby refining the diagnostic process, boosting accuracy, and increasing the rate of exclusion for severe pathologies.

The article is well-written, featuring concise language and a clear, logical structure.

The detailed explanation of the methodology, with all technical terms and concepts clearly defined and explained, along with the thorough description of the experiments conducted and the results obtained, facilitates a clear understanding of the presented material. The inclusion of visual aids, such as the diagram in Figure 1 and the use of pseudo-code to describe the three algorithms, further enhances reader comprehension.

The findings are effectively addressed, with a detailed comparison between the proposed algorithm and existing methods, including a discussion of strengths and weaknesses.

Additionally, the study’s limitations are presented, and potential directions for future research or improvements to the algorithm are outlined.

Here are some suggestions related to paper format and necessary improvements to enhance the quality of the paper:

·         Including a brief overview of the paper's content by chapters at the end of the Introduction would be useful.

·         Please review Figure 1 to ensure the following: (1) Each branch should have arrows; (2) It should be clear how to decide which path to follow if multiple options are available; (3) Define the meaning of W_ex and W_co (exploration and confirmation weights ?) and use consistent notations/abbreviations in Figure 1, Algorithm 3, and line 301.

·         Figure 1 is not referred in the text of the paper;

·         Sub-chapters 3.3 and 3.3.1 have the same content;

·         Sub-chapters 3.3.1 and 3.3.2 have the same title;

·         In the paper, only three equations are numbered (those under lines 163, 370, and 386), even though there are seven other unnumbered equations between equations (1) and (2). The authors are kindly requested to number all the equations.

·         References are required for the AARLC, Diaformer, BED, and CASANDE methods. (lines 401-402).

·         According to the guidelines of Entropy journal, references MUST be included at the end of the paper and cited within the text.

I hope my feedback is useful to the authors in improving their paper and wish them all the best in pursuing this important area of research.

Author Response

We sincerely appreciate the time and effort you have taken to review our manuscript. Your insightful comments have been invaluable in helping us refine and improve the quality of our work. Below, we provide detailed responses to each of your suggestions:

Comments 1: Including a brief overview of the paper's content by chapters at the end of the Introduction would be useful.

Response 1: Thank you for the suggestion. We have added a brief overview of each chapter at the end of the Introduction. Below is the revised content:

"This research presents an Adaptive Top-K Algorithm (TPKA) designed for medical conversational diagnostic systems. The related work section reviews existing telemedicine diagnostic systems and the application of the Top-K algorithm in the medical field, emphasizing the need for enhanced diagnostic accuracy through reinforcement learning. The methodology section details the design and implementation of TPKA, focusing on its dynamic adjustment mechanism, which optimizes the number of symptoms and diseases considered during diagnosis, thereby improving system efficiency and reducing computational overhead. The experiments validate TPKA's superior performance in differential diagnosis tasks compared to existing systems, demonstrating its effectiveness in enhancing diagnostic efficiency and handling severe pathologies. The discussion highlights TPKA's potential to improve diagnostic accuracy and efficiency in conversational systems, acknowledges the study's limitations, and suggests future research directions, such as using more representative datasets and integrating physician feedback to further optimize system performance. We believe this will enhance the clarity and accuracy of the paper."

The revised content can be found from pages 2 , lines 70 to 82.

Comments 2: Please review Figure 1 to ensure the following: (1) Each branch should have arrows; (2) It should be clear how to decide which path to follow if multiple options are available; (3) Define the meaning of W_ex and W_co (exploration and confirmation weights ?) and use consistent notations/abbreviations in Figure 1, Algorithm 3, and line 301.

Response 2: Thank you for your valuable suggestions. We have made the necessary adjustments to Figure 1. Arrows have been added to each branch, and we have clarified the decision-making process for paths with multiple options by providing distinct calculation methods. Additionally, we have defined the meanings of W_Ex and W_Co (exploration and confirmation weights) in the figure caption and ensured consistent notations/abbreviations throughout Figure 1, Algorithm 3, and line 301.

To further enhance the clarity of the figure, we have used different colors to distinguish between the different scopes of the Top-K algorithm: the path for calculating the Top-K symptoms is marked in blue, the path for calculating the Top-K pathologies is marked in yellow, and the comprehensive calculation path is marked in black. We believe this will enhance the clarity and accuracy of the paper.

The revised content can be found from pages 5 Figure 1.

Comments 3: Figure 1 is not referred in the text of the paper

Response 3: Thank you for your suggestion. In fact, Figure 1 is referenced and introduced in the text within section 3.1.1. Specifically, this reference can be found on page 4 line 183.

Comments 4: Sub-chapters 3.3 and 3.3.1 have the same content

Response 4: We sincerely apologize for the oversight that led to the retention of same content for sub-chapters 3.3.1 and 3.3.2. This was an unintended result of the drafting process. We have now deleted the content of sub-chapters 3.3.1, ensuring clarity and distinction in the paper. Thank you very much for bringing this to our attention.

Comments 5: Sub-chapters 3.3.1 and 3.3.2 have the same title

Response 5: We sincerely apologize for the oversight that led to the retention of same titles for sub-chapters 3.3.1 and 3.3.2. We have now deleted the title of sub-chapters 3.3.1, ensuring clarity and distinction in the paper. Thank you very much for bringing this to our attention.

Comments 6: In the paper, only three equations are numbered (those under lines 163, 370, and 386), even though there are seven other unnumbered equations between equations (1) and (2). The authors are kindly requested to number all the equations.

Response 6: Thank you for your valuable suggestion. We have now added numbers to all equations and cited them appropriately within the text. We believe these adjustments will improve the overall coherence and readability of the paper.

Comments 7: References are required for the AARLC, Diaformer, BED, and CASANDE methods. (lines 401-402).

Response 7: Thank you for bringing this to our attention. We apologize for the oversight. We have now added the appropriate references for the AARLC, Diaformer, BED, and CASANDE methods in the Experiments section. The revised content can be found from page 12 line 416-417.

Comments 8: According to the guidelines of Entropy journal, references MUST be included at the end of the paper and cited within the text.

Response 8: Thank you for the reminder. We have thoroughly reviewed the paper to ensure that all references are correctly cited within the text and included at the end of the paper in accordance with the guidelines of the Entropy journal. We believe this will enhance the clarity and accuracy of the paper.

Once again, we would like to express our gratitude for your constructive feedback. We believe that the revisions we have made in response to your comments have significantly strengthened our manuscript. We look forward to any further feedback you may have.

Reviewer 3 Report

Comments and Suggestions for Authors

Potential biases, such as data bias, confirmation bias, selection bias, algorithmic bias, and cognitive biases like the framing effect, which may impact the results, have not been explicitly discussed in the discussion section but have not been addressed in the methodology section.

what type of Encoder Network has been used in this algorithm? detailed description of the encode needs to be elaborated in the methodology section

is the algorithm utilizes a deep neural network architecture as part of its design? network architecture needs to be elaborate in the methodology section

Author Response

We sincerely appreciate the time and effort you have taken to review our manuscript. Your insightful comments have been invaluable in helping us refine and improve the quality of our work. Below, we provide detailed responses to each of your suggestions:

Comments 1: Potential biases, such as data bias, confirmation bias, selection bias, algorithmic bias, and cognitive biases like the framing effect, which may impact the results, have not been explicitly discussed in the discussion section but have not been addressed in the methodology section.

Response 1: Thank you for your insightful feedback. I have revised the paper to address the concerns regarding the discussion of potential biases. Specifically, I have added detailed discussions on data bias and algorithmic bias in the "Discussion" section. The revised content is as follows:

"Firstly, TPKA relies on the DDX Plus dataset for training and validation. Although this dataset contains detailed information on differential diagnoses and pathology severity, it may not fully capture the diversity of real-world medical scenarios, potentially leading to data bias. This bias could impair TPKA’s performance when dealing with diverse populations or rare conditions, reducing the system’s generalizability and robustness. Secondly, while TPKA’s dynamic adjustment mechanism optimizes resource allocation during the diagnostic process, it may also introduce algorithmic bias. The algorithm might prioritize based on historical data and common diseases, potentially overlooking atypical symptoms or rare conditions, thus increasing the risk of misdiagnosis or missed diagnosis. Furthermore, TPKA heavily depends on the quality of input data and real-time feedback for its dynamic adjustments. Although this dependency enhances the system’s flexibility and adaptability across various diagnostic stages, it also means that the system’s performance may be adversely affected in cases of low-quality input data or delayed feedback. Therefore, ensuring the quality of input data and the accuracy of real-time feedback is a critical challenge for future research. To address these biases and challenges, future studies should consider using more diverse and representative datasets and incorporating more sophisticated dynamic adjustment strategies to ensure that the algorithm is more sensitive when handling atypical symptoms and rare diseases."

The revised content can be found on page 15, lines 560-583.

These revisions ensure a comprehensive treatment of the potential biases that could affect the system’s performance, aligning the methodology and discussion sections with your observations.

Comments 2: what type of Encoder Network has been used in this algorithm? detailed description of the encode needs to be elaborated in the methodology section

Response 2: Thank you for pointing out the need for a more detailed description of the Encoder Network in the methodology section. I have revised the methodology to include a comprehensive explanation of the Encoder Network used in the system. The revised content is as follows:

"A key component is the Encoder Network, a multilayer perceptron (MLP) responsible for processing various data collected from the patient, including symptoms, medical history, and demographic information. The task of the Encoder Network is to transform these raw input data into a latent representation, a high-dimensional feature vector that captures the essential characteristics and complex relationships within the input data."

The revised content, including the detailed description of the Encoder Network, can be found in the methodology section from page 5, line 190-195.

Comments 3: is the algorithm utilizes a deep neural network architecture as part of its design? network architecture needs to be elaborate in the methodology section

Response 3: Thank you for your feedback regarding the need for a more detailed explanation of the deep neural network architecture utilized in the algorithm. I have revised the methodology section to provide a clearer overview of the network architecture employed in the system. The revised content is as follows:

"In this system, deep learning architectures are employed to achieve automated evidence acquisition and differential diagnosis, emulating the diagnostic process of a physician. The system's design draws from multiple prior studies, effectively integrating reinforcement learning and deep neural networks to assess and make decisions regarding a patient's condition. A key component is the Encoder Network, a multilayer perceptron (MLP) responsible for processing various data collected from the patient, including symptoms, medical history, and demographic information. The task of the Encoder Network is to transform these raw input data into a latent representation, a high-dimensional feature vector that captures the essential characteristics and complex relationships within the input data. The Classifier Network utilizes the latent representation generated by the Encoder Network to predict the differential diagnosis of the patient, inferring a list of potential diseases based on the current evidence. Meanwhile, the Policy Network leverages this latent representation to determine further actions, such as querying additional symptom information or terminating the interaction."

The revised content can be found in the methodology section from page 4, line 186 to page 5, line 199.

Once again, we would like to express our gratitude for your constructive feedback. We believe that the revisions we have made in response to your comments have significantly strengthened our manuscript. We look forward to any further feedback you may have.

Round 2

Reviewer 2 Report

Comments and Suggestions for Authors

I express my gratitude to the authors for considering all of my recommendations for enhancing the work.

To make the manuscript's final format better, I recommend the following:

 1. Authors are asked to include a space between the call of a figure and its number (e.g. Figure 1), of an equation (e.g. Equation 9) or when referring to a work included in References (e.g. AARLC [8]);

 2. I don't think the references for the four systems included in lines 416-417 should be reintroduced in lines 432-433.

Author Response

We appreciate your detailed review and valuable suggestions. We have carefully considered your feedback and made the following revisions:

Comments 1: Authors are asked to include a space between the call of a figure and its number (e.g. Figure 1), of an equation (e.g. Equation 9) or when referring to a work included in References (e.g. AARLC [8]);

Response 1: Thank you for pointing out the formatting issue. We have corrected it by ensuring that there is a space between the figure number (e.g., Figure 1), equation number (e.g., Equation 9), and references (e.g., AARLC [8]).

Comments 2: I don't think the references for the four systems included in lines 416-417 should be reintroduced in lines 432-433.

Response 2: We agree with your observation regarding the references for the four systems mentioned in lines 416-417. In line with your suggestion, we have revised the manuscript to avoid reintroducing these references in lines 432-433.

We are grateful for your corrections and believe that these improvements have enhanced the clarity and overall quality of the manuscript. Thank you once again for your valuable feedback.